# Optimized Dimensionality Reduction Methods for Interval-Valued Variables and Their Application to Facial Recognition

**DOI:** 10.3390/e21101016

**Published:** 2019-10-19

**Authors:** Jorge Arce Garro, Oldemar Rodríguez Rojas

**Affiliations:** 1National Bank of Costa Rica, 10101 San José, Costa Rica; jarceg@bncr.fi.cr; 2School of Mathematics, Research Center in Pure and Applied Mathematics (CIMPA), University of Costa Rica, 10101 San José, Costa Rica

**Keywords:** interval-valued variables, principal component analysis, symbolic data analysis, Best Point method

## Abstract

The center method, which was first proposed in *Rev. Stat. Appl.* 1997 by Cazes et al. and *Stat. Anal. Data Mining* 2011 by Douzal-Chouakria et al., extends the well-known Principal Component Analysis (PCA) method to particular types of symbolic objects that are characterized by multivalued interval-type variables. In contrast to classical data, symbolic data have internal variation. The authors who originally proposed the center method used the center of a hyper-rectangle in $R^{m}$ as a base point to carry out PCA, followed by the projection of all vertices of the hyper-rectangles as supplementary elements. Since these publications, the center point of the hyper-rectangle has typically been assumed to be the best point for the initial PCA. However, in this paper, we show that this is not always the case, if the aim is to maximize the variance of projections or minimize the squared distance between the vertices and their respective projections. Instead, we propose the use of an optimization algorithm that maximizes the variance of the projections (or that minimizes the distances between the squares of the vertices and their respective projections) and finds the optimal point for the initial PCA. The vertices of the hyper-rectangles are, then, projected as supplementary variables to this optimal point, which we call the “Best Point” for projection. For this purpose, we propose four new algorithms and two new theorems. The proposed methods and algorithms are illustrated using a data set comprised of measurements of facial characteristics from a study on facial recognition patterns for use in surveillance. The performance of our approach is compared with that of another procedure in the literature, and the results show that our symbolic analyses provide more accurate information. Our approach can be regarded as an optimization method, as it maximizes the explained variance or minimizes the squared distance between projections and the original points. In addition, the symbolic analyses generate more informative conclusions, compared with the classical analysis in which classical surrogates replace intervals. All the methods proposed in this paper can be executed in the RSDA package developed in R.

## 1. The Center Method

Symbolic data were introduced by Diday in [1].  In contrast to classical data analysis, in which a variable takes a single value, a variable in symbolic data can take a finite or infinite set of values: For example, an interval variable can take an infinite set of numerical values that range from low to high. As Principal Component Analysis (PCA) is one of the most popular multivariate methods for dimension reduction, its extension to symbolic data is important. Many generalizations of PCA have been developed and several studies have contributed to its extension to interval-valued data. Among the methods for this in the literature, two are the vertex method and the center method [2,3,4]. In [5], the authors introduced new PCA techniques in order to visualize and compare the structures of interval data. Then, the authors of [6] proposed an approach that extended the classical PCA method to interval-valued data by using symbolic covariance to determine the principal component space to reflect the total variation in the interval-valued data. PCA has also been extended to histogram data in a number of studies (see [7,8,9,10,11]).

Most of these methods were developed for interval matrices, where an interval matrix *X* is defined as
(1)$$X=\left[\begin{array}{cccc} \left[a_{11},b_{11}\right] & \left[a_{12},b_{12}\right] & \cdots & \left[a_{1m},b_{1m}\right]\\ \left[a_{21},b_{21}\right] & \left[a_{22},b_{22}\right] & \cdots & \left[a_{2m},b_{2m}\right]\\ \vdots & \vdots & \ddots & \vdots \\ \left[a_{n1},b_{n1}\right] & \left[a_{n2},b_{n2}\right] & \cdots & \left[a_{nm},b_{nm}\right] \end{array}\right],$$ where $a_{ij}\leq b_{ij}$ for all i=1,2,⋯,n and j=1,2,⋯,m (others authors denote the interval $\left[a_{ij},b_{ij}\right]$ by $\left[x_{ij}^{lo},x_{ij}^{up}\right]$ or $\left[\underline{x_{ij}},\bar{x_{ij}}\right]$ for all i=1,2,⋯,n and j=1,2,⋯,m). An interval matrix can be considered a subset of a matrix $M_{n\times m}$, which we denote by X, such that $X=\left\{Z\in M_{n\times m}|\forall i\in \left\{1,2,...,n\right\},\forall j\in \left\{1,2,...,m\right\},Z_{ij}\in \left[a_{ij},b_{ij}\right]\right\}$. In this case, Z∈X.

The center matrix of *X* is defined as
(2)$$X^{c}=\left[\begin{array}{cccc} X_{11}^{c} & X_{12}^{c} & \cdots & X_{1m}^{c}\\ X_{21}^{c} & X_{22}^{c} & \cdots & X_{2m}^{c}\\ \vdots & \vdots & \ddots & \vdots \\ X_{n1}^{c} & X_{n2}^{c} & \cdots & X_{nm}^{c} \end{array}\right],$$ where
(3)$$X_{ij}^{c}=\frac{a_{ij}+b_{ij}}{2}.$$

Here, $X^{c}\in X$ for i=1,⋯,n and j=1,⋯,m; $X_{ij}^{c}$ is a real number and not an interval. Thus, the center matrix (Equation 2) is a classical matrix. The center principal component method starts from the center matrix; in other words, classical PCA is applied to the center matrix $X^{c}$. Then, the *k*th principal components of the centers are
(4)$$Y_{\left(k\right)}^{c}=X^{c}v_{k}^{c},$$ where $v_{k}^{c}$ is the *k*th eigenvector of the variance–covariance matrix of $X^{c}$, defined in Equation (Equation 2). For cases (rows) i=1,⋯,n, the *k*th principal component for an interval variable is constructed as follows. Let $Y_{ik}^{c}=\left[y_{ik}^{lo},y_{ik}^{up}\right]$ be the interval principal component for an interval variable. Then,
(5)$$y_{ik}^{lo}=\sum_{j\in J_{c}^{-}}\left(b_{ij}-{\bar{X}}_{\left(j\right)}\right)v_{kj}^{c}+\sum_{j\in J_{c}^{+}}\left(a_{ij}-{\bar{X}}_{\left(j\right)}\right)v_{kj}^{c},$$
(6)$$y_{ik}^{up}=\sum_{j\in J_{c}^{-}}\left(a_{ij}-{\bar{X}}_{\left(j\right)}\right)v_{kj}^{c}+\sum_{j\in J_{c}^{+}}\left(b_{ij}-{\bar{X}}_{\left(j\right)}\right)v_{kj}^{c},$$ where $J_{c}^{-}=\left\{j|v_{kj}^{c}<0\right\}$ and $J_{c}^{+}=\left\{j|v_{kj}^{c}\geq 0\right\}$. More details can be found in [2].

The dual problem in the center PCA method was introduced by Rodriguez in [12]. To generalize duality relations, we let *D* be an interval matrix, defined as
$$D_{ij}=\left[\frac{a_{ij}-{\bar{X}}_{\left(j\right)}^{c}}{\sqrt{n}\sigma_{\left(j\right)}},\frac{b_{ij}-{\bar{X}}_{\left(j\right)}^{c}}{\sqrt{n}\sigma_{\left(j\right)}}\right],$$ for i=1,⋯,n and j=1,⋯,m, with ${\bar{X}}_{\left(j\right)}^{c}$ and $\sigma_{\left(j\right)}$ denoting the average and standard deviation of column *j*, respectively. The formulas shown in Theorem 1 are thus obtained and can be used to calculate the projections of interval variables.

**Theorem** **1.**
*If the hyper-rectangle defined by the jth column of D in the ith principal component is projected in the direction of $v_{i}$, then the minimum and maximum values can be computed by Equations (Equation 7) and (Equation 8), respectively.*
(7)$$\underline{r_{ij}}=\sum_{k=1,v_{kj}<0}^{m}{\bar{d}}_{ki}^{c}v_{kj}+\sum_{k=1,v_{kj}>0}^{m}{\underline{d}}_{ki}^{c}v_{kj},$$
(8)$$\bar{r_{ij}}=\sum_{k=1,v_{kj}<0}^{m}{\underline{d}}_{ki}^{c}v_{kj}+\sum_{k=1,v_{kj}>0}^{m}{\bar{d}}_{ki}^{c}v_{kj}.$$


The proof of this theorem can be found in [12,13].

## 2. The Best Point Method

Let *X* be an n×m matrix of interval variables and let Z∈X. If we apply PCA to a matrix *Z*, then the *k*th principal component of *Z* for an observation $\xi_{u}$, with k=1,⋯,s<mandi=1,⋯,m, is
(9)$$y_{ik}^{Z}=\sum_{j=1}^{m}\left(Z_{jk}-{\bar{Z}}_{\left(j\right)}\right)w_{k_{j}}^{Z},$$ where ${\bar{Z}}_{\left(j\right)}$ is the average of the variable $Z_{\left(j\right)}$ (i.e., ${\bar{Z}}_{\left(j\right)}=\frac{1}{n}\sum_{i=1}^{n}Z_{ij}$), and $w_{k}^{Z}=\left(w_{k_{1}}^{Z},\cdots ,w_{k_{m}}^{Z}\right)$ is the *k*th eigenvector associated with the variance–covariance matrix of *Z*. It is clear that $\beta \left(Z\right)=\left\{w_{1}^{Z},\cdots ,w_{m}^{Z}\right\}$ is an orthonormal basis of $R^{m}$.

For the matrix *X* defined in (Equation 1), we let $X_{i}=\left(\left[a_{i1},b_{i1}\right],\cdots ,\left[a_{im},b_{im}\right]\right)$ for i=1,⋯,n. Then, we define the vertex matrix for an observation *i* as
(10)$$X_{i}^{v}=\left[\begin{array}{cccc} a_{i1} & a_{i2} & \cdots & a_{im}\\ a_{i1} & a_{i2} & \cdots & b_{im}\\ \vdots & \vdots & \ddots & \vdots \\ b_{i1} & b_{i2} & \cdots & a_{im}\\ b_{i1} & b_{i2} & \cdots & b_{im} \end{array}\right].$$

Thus, the vertex matrix of *X* is
(11)$$X^{v}=\left[\begin{array}{c} \left[\begin{array}{cccc} a_{11} & a_{12} & \cdots & a_{1m}\\ \vdots & \vdots & \ddots & \vdots \\ b_{11} & b_{12} & \cdots & b_{1m} \end{array}\right]\\ \begin{array}{cccc} \vdots & \vdots & \vdots & \vdots \end{array}\\ \left[\begin{array}{cccc} a_{i1} & a_{i2} & \cdots & a_{im}\\ \vdots & \vdots & \ddots & \vdots \\ b_{i1} & b_{i2} & \cdots & b_{im} \end{array}\right]\\ \begin{array}{cccc} \vdots & \vdots & \vdots & \vdots \end{array}\\ \left[\begin{array}{cccc} a_{n1} & a_{n2} & \cdots & a_{nm}\\ \vdots & \vdots & \ddots & \vdots \\ b_{n1} & b_{n2} & \cdots & b_{nm} \end{array}\right] \end{array}\right].$$

Next, the rows of the vertex matrix of *X* are projected as supplementary elements in the PCA of *Z*. We define the supplementary vertex matrix as
(12)$${\tilde{X}}_{i}^{v}\left(Z\right)=\left[\begin{array}{cccc} \frac{a_{i1}-{\bar{Z}}_{\left(1\right)}}{\sqrt{n}\sigma_{\left(1\right)}} & \frac{a_{i2}-{\bar{Z}}_{\left(2\right)}}{\sqrt{n}\sigma_{\left(2\right)}} & \cdots & \frac{a_{im}-{\bar{Z}}_{\left(m\right)}}{\sqrt{n}\sigma_{\left(m\right)}}\\ \frac{a_{i1}-{\bar{Z}}_{\left(1\right)}}{\sqrt{n}\sigma_{\left(1\right)}} & \frac{a_{i2}-{\bar{Z}}_{2}}{\sqrt{n}\sigma_{\left(2\right)}} & \cdots & \frac{a_{im}-{\bar{Z}}_{\left(m\right)}}{\sqrt{n}\sigma_{\left(m\right)}}\\ \vdots & \vdots & \ddots & \vdots \\ \frac{b_{i1}-{\bar{Z}}_{\left(1\right)}}{\sqrt{n}\sigma_{\left(1\right)}} & \frac{b_{i2}-{\bar{Z}}_{\left(2\right)}}{\sqrt{n}\sigma_{\left(2\right)}} & \cdots & \frac{b_{im}-{\bar{Z}}_{\left(m\right)}}{\sqrt{n}\sigma_{\left(m\right)}} \end{array}\right],$$ where $\sigma_{\left(j\right)}$ is the standard deviation of $Z_{\left(j\right)}$. To simplify this approach, we denote each row of the matrix ${\tilde{X}}_{i}^{v}\left(Z\right)$ by ${\tilde{x}}_{i_{tj}}^{v}\left(Z\right),$ with $t=1,\cdots ,2^{m_{i}}$, in which $m_{i}$ is the number of nontrivial intervals, and j=1,⋯,m. Then, the co-ordinates are obtained:(13)$$C^{k}\left(x_{i_{j}}^{v}\right)=\sum_{h=1}^{m}{\tilde{x}}_{i_{jh}}^{v}\left(Z\right)w_{k_{h}},$$ with $j=1,\cdots ,2^{m_{i}}$, in which $m_{i}$ is the number of nontrivial intervals. Then, the minimum and maximum of the interval can be calculated:(14)$${\tilde{Y}}_{ik}^{v}={\tilde{y}}_{ik}=\left[{\tilde{y}}_{ik}^{a_{Z}},{\tilde{y}}_{ik}^{b_{Z}}\right]\;\mathrm{with}\;k=1,\cdots ,s<m,$$
and
(15)$${\tilde{y}}_{ik}^{a_{Z}}=\left[j=1,\cdots ,2^{m_{i}}\right]\min C^{k}\left(x_{i_{j}}^{v}\right),$$
(16)$${\tilde{y}}_{ik}^{b_{Z}}=\left[j=1,\cdots ,2^{m_{i}}\right]\max C^{k}\left(x_{i_{j}}^{v}\right).$$

The formulas in the following theorem allow us to compute Equations (Equation 15) and (Equation 16) much more quickly.

**Theorem** **2.**
*The co-ordinates of ${\tilde{Y}}_{ik}^{v_{Z}}$ can be found as follows:*
$${\tilde{y}}_{ik}^{a_{Z}}=\sum_{j\in J_{Z}^{-}}\left(b_{ij}-{\bar{Z}}_{\left(j\right)}\right)w_{k_{j}}^{Z}+\sum_{j\in J_{Z}^{+}}\left(a_{ij}-{\bar{Z}}_{\left(j\right)}\right)w_{k_{j}}^{Z},$$
$${\tilde{y}}_{ik}^{b_{Z}}=\sum_{j\in J_{Z}^{-}}\left(a_{ij}-{\bar{Z}}_{\left(j\right)}\right)w_{k_{j}}^{Z}+\sum_{j\in J_{Z}^{+}}\left(b_{ij}-{\bar{Z}}_{\left(j\right)}\right)w_{k_{j}}^{Z},$$
*where $J_{Z}^{-}=\left\{j|w_{kj}^{v}<0\right\}$, $J_{Z}^{+}=\left\{j|w_{kj}^{v}\geq 0\right\}$, and ${\bar{Z}}_{\left(j\right)}$ is the mean of the jth column.*


**Proof.** Let Z∈X; then,
(17)$$\forall j,\;\forall i,\;Z_{ij}\in X_{ij}=\left[a_{ij},b_{ij}\right].$$As $a_{ij}$ and $b_{ij}$ are supplementary elements in the PCA of *Z*, they must first be centered with respect to the columns (variables) of *Z*. Thus,
(18)$$\forall i,\;\forall j,\;a_{ij}-{\bar{Z}}_{\left(j\right)},\;b_{ij}-{\bar{Z}}_{\left(j\right)}.$$Then, from Equations (Equation 17) and (Equation 18),
(19)$$\forall i,\;\forall j,\;z_{ij}-{\bar{Z}}_{\left(j\right)}\in X_{ij}-{\bar{Z}}_{\left(j\right)}=\left[a_{ij}-{\bar{Z}}_{\left(j\right)},b_{ij}-{\bar{Z}}_{\left(j\right)}\right].$$
**Case 1:**$\forall j,\;w_{k_{j}}^{Z}>0,$$$\forall j,\;\forall i,\;w_{k_{j}}^{Z}\left(z_{ij}-{\bar{Z}}_{\left(j\right)}\right)\in w_{k_{j}}^{Z}\left[a_{ij}-{\bar{Z}}_{\left(j\right)},b_{ij}-{\bar{Z}}_{\left(j\right)}\right]\;\Rightarrow \;\sum_{j=1}^{p}\left(a_{ij}-{\bar{Z}}_{\left(j\right)}\right)w_{k_{j}}^{Z}\leq \sum_{j=1}^{p}\left(z_{ij}-{\bar{Z}}_{\left(j\right)}\right)w_{k_{j}}^{Z}\leq \sum_{j=1}^{p}\left(b_{ij}-{\bar{Z}}_{\left(j\right)}\right)w_{k_{j}}^{Z}.$$**Case 2:**$\forall j,\;w_{kj}^{Z}<0,$$$\forall j,\;\forall i,\;w_{k_{j}}^{Z}\left(z_{ij}-{\bar{Z}}_{\left(j\right)}\right)\in w_{k_{j}}^{Z}\left[b_{ij}-{\bar{Z}}_{\left(j\right)},a_{ij}-{\bar{Z}}_{\left(j\right)}\right]\;\Rightarrow \;\sum_{j=1}^{p}\left(b_{ij}-{\bar{Z}}_{\left(j\right)}\right)w_{k_{j}}^{Z}\leq \sum_{j=1}^{p}\left(z_{ij}-{\bar{Z}}_{\left(j\right)}\right)w_{k_{j}}^{Z}\leq \sum_{j=1}^{p}\left(a_{ij}-{\bar{Z}}_{\left(j\right)}\right)w_{k_{j}}^{Z}.$$**Case 3:** Let $J_{Z}^{-}=\left\{j|w_{k_{j}}<0\right\}$ and $J_{Z}^{+}=\left\{j|w_{k_{j}}\geq 0\right\}$.For **Case 1** applied to $J_{Z}^{+},$
(20)$$\sum_{j\in J_{Z}^{+}}\left(a_{ij}-{\bar{Z}}_{\left(j\right)}\right)w_{k_{j}}^{Z}\leq \sum_{j\in J_{Z}^{+}}\left(z_{ij}-{\bar{Z}}_{\left(j\right)}\right)w_{k_{j}}^{Z}\leq \sum_{j\in J_{Z}^{+}}\left(b_{ij}-{\bar{Z}}_{\left(j\right)}\right)w_{k_{j}}^{Z}.$$For **Case 2** applied to $J_{v}^{-},$
(21)$$\sum_{j\in J_{Z}^{+}}\left(b_{ij}-{\bar{Z}}_{\left(j\right)}\right)w_{k_{j}}^{Z}\leq \sum_{j\in J_{Z}^{+}}\left(z_{ij}-{\bar{Z}}_{\left(j\right)}\right)w_{k_{j}}^{Z}\leq \sum_{j\in J_{Z}^{+}}\left(a_{ij}-{\bar{Z}}_{\left(j\right)}\right)w_{k_{j}}^{Z}.$$Therefore, from Equations (Equation 20) and (Equation 21), we obtain
$$\begin{array}{cc} & \sum_{j\in J_{Z}^{-}}\left(b_{ij}-{\bar{Z}}_{\left(j\right)}\right)w_{k_{j}}^{Z}+\sum_{j\in J_{Z}^{+}}\left(a_{ij}-{\bar{Z}}_{\left(j\right)}\right)w_{k_{j}}^{Z}\\ \leq & \sum_{j=1}^{n}\left(z_{ij}-{\bar{Z}}_{\left(j\right)}\right)w_{k_{j}}^{Z}\\ \leq & \sum_{j\in J_{Z}^{-}}\left(a_{ij}-{\bar{Z}}_{\left(j\right)}\right)w_{k_{j}}^{Z}+\sum_{j\in J_{Z}^{+}}\left(b_{ij}-{\bar{Z}}_{\left(j\right)}\right)w_{k_{j}}^{Z}. \end{array}$$
Therefore,
$${\tilde{y}}_{ik}^{a_{Z}}=\sum_{j\in J_{Z}^{-}}\left(b_{ij}-{\bar{Z}}_{\left(j\right)}\right)w_{k_{j}}^{Z}+\sum_{j\in J_{Z}^{+}}\left(a_{ij}-{\bar{Z}}_{\left(j\right)}\right)w_{k_{j}}^{Z},$$
$${\tilde{y}}_{ik}^{b_{Z}}=\sum_{j\in J_{Z}^{-}}\left(a_{ij}-{\bar{Z}}_{\left(j\right)}\right)w_{k_{j}}^{Z}+\sum_{j\in J_{Z}^{+}}\left(b_{ij}-{\bar{Z}}_{\left(j\right)}\right)w_{k_{j}}^{Z}.$$□

The following theorem provides the co-ordinates in the variable space; this is a dual relationship. We need to center and standardize the matrix *Z*.
$${\tilde{Z}}_{ij}=\frac{Z_{ij}-{\bar{Z}}_{\left(j\right)}}{\sqrt{n}\sigma_{\left(j\right)}}.$$

Next, we next focus on the matrix $\tilde{Z}={\tilde{Z}}_{ij}$∀i, ∀j. Let ${\tilde{z}}^{j}$ be the *j*th column of $\tilde{Z}$, with ${\left({\tilde{z}}^{j}\right)}^{t}\cdot {\tilde{z}}^{i}=R\left(i,j\right)\leq 1$. Then, the interval matrix is centered and standardized with respect to *Z*:(22)$$\tilde{X}\left(Z\right)=\left[\begin{array}{cccc} \left[\frac{a_{11}-{\bar{Z}}_{\left(1\right)}}{\sqrt{n}\sigma_{\left(1\right)}},\frac{b_{11}-{\bar{Z}}_{\left(1\right)}}{\sqrt{n}\sigma_{\left(1\right)}}\right] & \left[\frac{a_{12}-{\bar{Z}}_{\left(2\right)}}{\sqrt{n}\sigma_{\left(2\right)}},\frac{b_{12}-{\bar{Z}}_{\left(2\right)}}{\sqrt{n}\sigma_{\left(2\right)}}\right] & \cdots & \left[\frac{a_{1m}-{\bar{Z}}_{\left(m\right)}}{\sqrt{n}\sigma_{\left(m\right)}},\frac{b_{1m}-{\bar{Z}}_{\left(m\right)}}{\sqrt{n}\sigma_{\left(m\right)}}\right]\\ \vdots & \vdots & \ddots & \vdots \\ \left[\frac{a_{i1}-{\bar{Z}}_{\left(1\right)}}{\sqrt{n}\sigma_{\left(1\right)}},\frac{b_{i1}-{\bar{Z}}_{\left(1\right)}}{\sqrt{n}\sigma_{\left(1\right)}}\right] & \left[\frac{a_{i2}-{\bar{Z}}_{\left(2\right)}}{\sqrt{n}\sigma_{\left(2\right)}},\frac{b_{i2}-{\bar{Z}}_{\left(2\right)}}{\sqrt{n}\sigma_{\left(2\right)}}\right] & \cdots & \left[\frac{a_{im}-{\bar{Z}}_{\left(m\right)}}{\sqrt{n}\sigma_{\left(m\right)}},\frac{b_{im}-{\bar{Z}}_{\left(m\right)}}{\sqrt{n}\sigma_{\left(m\right)}}\right]\\ \vdots & \vdots & \ddots & \vdots \\ \left[\frac{a_{n1}-{\bar{Z}}_{\left(1\right)}}{\sqrt{n}\sigma_{\left(1\right)}},\frac{b_{n1}-{\bar{Z}}_{\left(1\right)}}{\sqrt{n}\sigma_{\left(1\right)}}\right] & \left[\frac{a_{n2}-{\bar{Z}}_{\left(2\right)}}{\sqrt{n}\sigma_{\left(2\right)}},\frac{b_{n2}-{\bar{Z}}_{\left(2\right)}}{\sqrt{n}\sigma_{\left(2\right)}}\right] & \cdots & \left[\frac{a_{nm}-{\bar{Z}}_{\left(m\right)}}{\sqrt{n}\sigma_{\left(m\right)}},\frac{b_{nm}-{\bar{Z}}_{\left(m\right)}}{\sqrt{n}\sigma_{\left(m\right)}}\right] \end{array}\right].$$

To facilitate the analysis, we define
$${\left(\tilde{X}\left(Z\right)\right)}_{ij}=\left[\frac{a_{ij}-{\bar{Z}}_{\left(j\right)}}{\sqrt{n}\sigma_{\left(j\right)}},\frac{b_{ij}-{\bar{Z}}_{\left(j\right)}}{\sqrt{n}\sigma_{\left(j\right)}}\right]=\left[a_{ij}^{Z},b_{ij}^{Z}\right].$$

The inertia matrix $\tilde{Z}{\tilde{Z}}^{t}$ is symmetric and positive semidefinite, so all its eigenvectors are orthogonal and its eigenvalues are real and nonnegative. We let $v_{1}^{Z},v_{2}^{Z},\cdots ,v_{s}^{Z}$ denote the *s* eigenvectors of $\tilde{Z}{\tilde{Z}}^{t}$ associated with eigenvalues $\lambda_{1},\lambda_{2},\cdots ,\lambda_{s}\geq 0$. Then, $V\left(Z\right)=\left[v_{1}^{Z}|v_{2}^{Z}\left|\cdots \right|v_{s}^{Z}\right]$ is defined as a matrix of the size n×s whose columns are the eigenvectors of $\tilde{Z}{\tilde{Z}}^{t}$. We can compute the co-ordinates of the variables in the correlation circle as ${\tilde{Z}}^{t}V$, and we can then compute the *i*th column of *Z* in the *j*th principal component (in the $v_{j}^{Z}$ direction) using Equation (Equation 23).
(23)$$r_{ij}^{Z}=\sum_{k=1}^{m}{\tilde{Z}}_{ki}v_{kj}^{Z}.$$

The next theorem proves the duality relation of any matrix that belongs to an interval matrix.

**Theorem** **3.**
*If the hyper-rectangle defined by the jth column of $\tilde{X}\left(Z\right)$ in the ith principal component is projected in the direction of $v_{i}$, then the maximum and minimum values can be obtained by Equations (Equation 24) and (Equation 25), respectively.*
(24)$$\underline{r_{ij}}=\sum_{k=1,v_{kj}<0}^{m}b_{ki}^{Z}v_{kj}+\sum_{k=1,v_{kj}>0}^{m}a_{ki}^{Z}v_{kj},$$
(25)$$\bar{r_{ij}}=\sum_{k=1,v_{kj}<0}^{m}a_{ki}^{Z}v_{kj}+\sum_{k=1,v_{kj}>0}^{m}b_{ki}^{Z}v_{kj}.$$


**Proof.** The proof of this theorem is similar to the proof of Theorem 2. □

In the above, we prove that $p{\hat{z}}_{ij}\in \left[\underline{r_{ij}},\bar{r_{ij}}\right]$ and that $\underline{r_{ij}}$ and $\bar{r_{ij}}$ are a combination of the projections of the vertices of the hyper-rectangle $R^{m}$. We can form duality relations between the eigenvectors of $\tilde{Z}{\tilde{Z}}^{t}$ and ${\tilde{Z}}^{t}\tilde{Z}$: Both matrices have the same *s* positive eigenvalues $\lambda_{1}^{Z},\lambda_{2}^{Z},\ldots ,\lambda_{s}^{Z}$, and if $u_{1}^{Z},u_{2}^{Z},\ldots ,u_{s}^{Z}$ are the first *s* eigenvectors of ${\tilde{Z}}^{t}\tilde{Z}$, then the relations between the eigenvectors of $\tilde{Z}{\tilde{Z}}^{t}$ and ${\tilde{Z}}^{t}\tilde{Z}$ can be computed by Equations (Equation 26) and (Equation 27):(26)$$u_{\ell }^{Z}=\frac{{\tilde{Z}}^{t}v_{\ell }^{Z}}{\sqrt{\lambda_{\ell }}}\;\mathrm{for}\;\ell =1,2,\ldots ,s.$$
(27)$$v_{\ell }^{Z}=\frac{\tilde{Z}u_{\ell }^{Z}}{\sqrt{\lambda_{\ell }}}\;\mathrm{for}\;\ell =1,2,\ldots ,s.$$

Above, we provide the theory to apply PCA to all matrices Z∈X. Now, we aim to find a matrix $Z^{*}\in X$ that is optimal for one of two criteria: (1) The minimization of the square of the distance from the vertices of the hypercubes to the principal axes of *Z*, or (2) the maximization of the variance of the first components of *Z*. We develop these concepts in the following two sections.

### 2.1. Minimizing the Square of the Distance from the Hypercube Vertices to the Principal Axes of *Z*

Let *X* be an interval of an n×m matrix, Z∈X, and $\beta \left(Z\right)=\left\{w_{1}^{Z},\cdots ,w_{s}^{Z}\right\}$, with s≤m and $w_{i}^{Z}$ eigenvectors of the variance–covariance matrix of *Z*. We let $X^{v}$ denote the vertex matrix of *X* and $N=\sum_{i=1}^{n}2^{m_{i}}$, in which $m_{i}$ is the number of nontrivial intervals for case $\xi_{i}$. Then, the centered and standardized vertex matrix with respect to *Z* has the following form:(28)$${\tilde{X}}^{v}\left(Z\right)=\left[\begin{array}{c} \left[\begin{array}{cccc} \frac{a_{11}-{\bar{Z}}_{\left(1\right)}}{\sqrt{n}\sigma_{\left(1\right)}} & \frac{a_{12}-{\bar{Z}}_{\left(2\right)}}{\sqrt{n}\sigma_{\left(2\right)}} & \cdots & \frac{a_{1m}-{\bar{Z}}_{\left(m\right)}}{\sigma_{\left(m\right)}}\\ \vdots & \vdots & \vdots & \vdots \\ \frac{b_{11}-{\bar{Z}}_{\left(1\right)}}{\sqrt{n}\sigma_{\left(1\right)}} & \frac{b_{12}-{\bar{Z}}_{\left(2\right)}}{\sqrt{n}\sigma_{\left(2\right)}} & \cdots & \frac{b_{1m}-{\bar{Z}}_{\left(m\right)}}{\sqrt{n}\sigma_{\left(m\right)}} \end{array}\right]\\ \begin{array}{cccc} \vdots & \vdots & \vdots & \vdots \end{array}\\ \left[\begin{array}{cccc} \frac{a_{n1}-{\bar{Z}}_{\left(1\right)}}{\sqrt{n}\sigma_{\left(1\right)}} & \frac{a_{n2}-{\bar{Z}}_{\left(2\right)}}{\sqrt{n}\sigma_{\left(2\right)}} & \cdots & \frac{a_{nm}-{\bar{Z}}_{\left(m\right)}}{\sqrt{n}\sigma_{\left(m\right)}}\\ \vdots & \vdots & \vdots & \vdots \\ \frac{b_{n1}-{\bar{Z}}_{\left(1\right)}}{\sqrt{n}\sigma_{\left(1\right)}} & \frac{b_{n2}-{\bar{Z}}_{\left(2\right)}}{\sqrt{n}\sigma_{\left(2\right)}} & \cdots & \frac{b_{nm}-{\bar{Z}}_{\left(m\right)}}{\sqrt{n}\sigma_{\left(m\right)}} \end{array}\right] \end{array}\right].$$

Let $\varphi \left(Z\right):X\to R^{+}\cup \left\{0\right\}$ be the function defined by
(29)$$\varphi \left(Z\right)=\sum_{i=1}^{N}{\tilde{X_{i}^{v}}\left(Z\right)-Pr_{\beta (Z)}\left(\tilde{X_{i}^{v}}\left(Z\right)\right)}^{2},$$ where ||.|| is the Euclidean norm. To compute φ(Z), we propose Algorithm 1:

**Algorithm 1** The computation of φ.
**Require:**
*X*
an n×m matrix of intervals, Z∈X, *s* number of principal components.  

**Ensure:**
φ(Z).  

1:
Apply PCA to *Z*.  
2:
$\beta =\{w_{1},\cdots ,w_{s}\}$, with s≤m and $w_{i}$ eigenvectors of the variance–covariance matrix of *Z*.  
3:
Compute the vertex matrix of *X*$\left(X^{v}\right)$.  
4:
Compute the vertex matrix of the centered and standardized *X* with respect to *Z*$\left(\tilde{X^{v}}\left(Z\right)\right)$.
5:
$\varphi \left(Z\right)=\sum_{i=1}^{N}{\tilde{X_{i}^{v}}\left(Z\right)-Pr_{\beta (Z)}\left(\tilde{X_{i}^{v}}\left(Z\right)\right)}^{2}$.  
6:
**return**
φ(Z).  



As Z∈X, X is a finite union of compact sets and φ(Z) is a continuous function, φ always reaches the minimum and the maximum. In this case, the aim is to obtain the matrix *Z* that minimizes the distance to the vertex matrix $X^{v}$. The problem that we aim to solve is
(30)$$\begin{array}{cc} \mathrm{Minimize} & \varphi \left(Z\right)=\sum_{i=1}^{N}{||{\tilde{X}}_{i}^{v}\left(Z\right)-Pr_{\beta (Z)}\left({\tilde{X}}_{i}^{v}\left(Z\right)\right)||}^{2}\\ \mathrm{Subject}\;\mathrm{to} & Z\in X. \end{array}$$

**Definition** **1.**
*The matrix Z∈X that solves Problem Equation 30 is the optimal matrix with respect to distance, which is denoted by $Z^{\varphi }$.*


To perform the optimization that computes $Z^{\varphi }$, we propose Algorithm 2:

**Algorithm 2** Computation of the Best Matrix with respect to the distances of the vertices.
**Require:**
*X*
a symbolic matrix of intervals of dimension n×m, Z∈X, *s* number of principal components, TOL is the variation tolerance between iterations, and *N* is the maximum number of iterations.  

**Ensure:**
$\tilde{Y^{V_{Z^{\varphi }}}}$.  

1:
Consider $Z=X^{c}$, the center matrix Equation 2, to be the initial value.  
2:
Get $Z^{\varphi }$ by means of optimization algorithm $\left(\mathrm{initialvalue}=Z,\mathrm{function}=\varphi (Z),TOL,N\right)$.
3:
Get $\tilde{Y^{V_{Z^{\varphi }}}}$ Use Theorem 2.  
4:
**return**
$\tilde{Y^{V_{Z^{\varphi }}}}$.  



### 2.2. Maximizing the Variance of the First Components

Let *X* be an interval matrix of dimension n×m, Z∈X, and $\beta \left(Z\right)=\left\{w_{1}^{Z},\cdots ,w_{s}^{Z}\right\}$, with s≤m and $w_{i}^{Z}$ eigenvectors of the variance–covariance matrix of *Z* and $\lambda \left(Z\right)=\lambda_{1}^{Z},\cdots ,\lambda_{s}^{Z}$ denoting the set of associated eigenvalues of the variance–covariance matrix of *Z*. We define the function $\Lambda \left(Z,s\right):X\times N\to R^{+}$ as $\Lambda \left(Z,s\right)=\sum_{i=1}^{s}\lambda_{i}^{Z}$. To compute Λ(Z,s), we propose Algorithm 3:

**Algorithm 3** The computation of Λ.
**Require:**
*X*
an n×m symbolic matrix of intervals of dimension Z∈X, *s* number of principal components.  

**Ensure:**
Λ(Z,s).  

1:
Apply PCA to *Z*.  
2:
$\lambda \left(Z\right)=\lambda_{1}^{Z},\cdots ,\lambda_{s}^{Z}$ set of associated eigenvalues of the variance–covariance matrix of *Z*.  
3:
$\Lambda \left(Z,s\right)=\sum_{i=1}^{s}\lambda_{i}^{Z}$.  
4:
**return**
Λ(Z,s).  



As above, since Z∈X and X is the finite union of compact sets with *s* number of principal components, Λ(Z,s) is a continuous function and, thus, always reaches the minimum and the maximum. In this case, the aim is to obtain the matrix *Z* that maximizes the accumulated inertia in the first *s* principal components. The problem that we want to solve is
(31)$$\begin{array}{cc} \mathrm{Maximize} & \Lambda \left(Z,s\right)=\sum_{i=1}^{s}\lambda_{i}^{Z}\\ \mathrm{Subject}\;\mathrm{to} & Z\in X. \end{array}$$

**Definition** **2.**
*The matrix Z∈X that solves Problem Equation 31 is the optimal matrix with respect to inertia, denoted by $Z^{\Lambda }$.*


To perform the optimization that computes $Z^{\Lambda }$, we propose Algorithm 4:

**Algorithm 4** The computation of the Best Matrix with respect to inertia.
**Require:**
*X*
an n×m symbolic matrix of intervals of dimension, Z∈X, *s* number of principal components.  

**Ensure:**
${\tilde{Y}}^{V_{Z^{\Lambda }}}$.  

1:
Consider $Z=X^{c}$, center matrix Equation 2, as the initial value.  
2:
Get $Z^{\Lambda }$ by means of the optimization algorithm $\left(\mathrm{initialvalue}=Z,\mathrm{function}=\Lambda (Z,s)\right)$.
3:
Get ${\tilde{Y}}^{V_{Z^{\Lambda }}}$ using Theorem 2.  
4:
**return**
$\tilde{Y^{V_{Z^{\Lambda }}}}$.  



## 3. Experimental Evaluation: The Application to Facial Recognition

Automatic facial recognition has recently gained momentum, especially in the context of security issues such as access to buildings, and in the context of monitoring and continued surveillance. A well-known application of facial recognition is its incorporation in the iPhone X. According to Apple’s support website, the technology that enables facial ID is some of the most advanced hardware and software that has ever been created. The TrueDepth camera captures accurate facial data by projecting and analyzing over 30,000 invisible dots to create a facial depth map, while also capturing an infrared image. These images are transformed into a mathematical representation, which is compared with registered facial data. In both R and Python, a significant number of libraries, such as the videoplayR package, have also been developed for facial recognition. The link below contains more details: http://www.stoltzmaniac.com/facial-recognition-in-r/.

As described in this section, we applied all the proposed methods using p=6 interval-valued variables in a data set of m=27 faces for a total of 27,000 photos. The data set was taken from [14], in which the authors investigated facial characteristics for detection purposes in a surveillance study. In this study, the center PCA method in [4] was applied, as shown in Figure 1. The data are provided in Table 1.

The data set contains measurements of p=6 random variables designed to identify each face: The length spanned by the eyes $X_{1}$ (distance AD in Figure 1), the length between the eyes $X_{2}$ (distance BC), the length from the outer right eye to the upper-middle lip at point H between the nose and mouth $X_{3}$ (distance AH), the corresponding length for the left eye $X_{4}$ (DH), the length from point H to the outside of the mouth on the right side $X_{5}$ (EH), and the corresponding distance to the left side of the mouth $X_{6}$ (GH). For each facial image in this facial recognition process, salient features, such as the nose, mouth, and eyes, are located using morphological operators. The boundaries of the located elements are extracted by using a specific active contour method based on Fourier descriptors, which incorporates information about the global shape of each object. Finally, the specific points delimiting each extracted boundary are located, and the distance is measured between a specific pair of points, as represented by the random variables in Figure 1. This distance measure is expressed as the number of pixels in a facial image. As there is a sequence of such images, the actual measured distances are interval-valued variables. Thus, for example, the eye span distance $X_{1}$ for case HUS1 is $X_{1}=\left[168.86,172.84\right]$ for this series of images. Notably, different conditions of alignment, illumination, pose, and occlusion cause variation in the distances extracted from different images of the same person. The study that generated the data set involved nine men and three sequences for each subject for a total of m=27 cases. The complete data set is provided in Table 1.

It is important to note that the data in Table 1 are aggregated. There are 27 interval-valued cases; if each case is drawn from a sequence of 1000 images, then there are 27,000 classical point observations in $R^{6}$. An underlying assumption of the standard classical analysis is that all 27,000 observations are independent. However, this is not the case in this data set. The data values for each face form a set of 1000 dependent observations. Therefore, if we were to use each image as a statistical unit by performing classical analysis, then we would lose information about the dependence contained in the 27,000 observations. The resulting principal component analysis would look for axes that maximize the variability across all 27,000 images, regardless of whether some images belong to the same sequence. In contrast, as interval-valued observations are obtained from each sequence, the Best Point method extracts the principal component axes that maximize the variability in each interval (i.e., those that maximize the internal variability), thereby retaining information on the dependency among the 1000 images in each sequence.

### Comparison between the Center, Vertex, and Best Point Methods

We applied the vertex, center, and best point principal component methods to the data in Table 1. The Best Point principal component method was run with two different goals: (1) To minimize the squared distance and (2) to maximize the variance. From this point, the Best Point principal component method that minimizes the squared distance was designated as the Best Point Distance, and the Best Point principal component method that maximizes the variance was designated as the Best Point Variance. Table 2 shows the first two principal components generated by the four methods.

Figure 2 compares the data in Table 1 with the principal planes generated by the vertex, center, Best Point Distance, and Best Point Variance principal component methods. The plots of these, along with the first principal component (PC1) and second principal component (PC2) axes, are shown in Figure 2. The proximity of the three sequences for the three faces for each individual can be readily observed, which validates their within-subject coherence. Furthermore, with all four methods, the same four classes of faces were distinguished. Faces {INC, FRA} can be regarded as one class, faces {HUS, PHI, JPL} and {ISA, ROM} constitute two other classes, and faces {LOT, KHA} form the fourth class. Of the four principal planes in the graph, those corresponding to the Best Point Distance and Best Point Variance methods show much better separation of the face classes, which results in superior facial classification. In other words, the proposed methods more accurately predicted the individual in the photo.

Numerical analysis results confirm that the separation of classes from the Best Point Distance and Best Point Variance methods was much better than that from the other methods. Table 3 compares the accumulated variance of the vertex, center, Best Point Distance, and Best Point Variance principal component methods. The better methods are clearly Best Point Distance and Best Point Variance, which, in the third principal component, reached 91.25% and 99.72% of the accumulated variance, respectively; both of which were far superior to the results of the center and vertex methods.

As shown in Table 4, for the criterion of the minimum distance between the corners of the original hyper-rectangles in $R^{m}$ and the principal components, the Best Point Distance method outperformed the other methods, with a minimum distance of 6676.43. This distance was significantly less than the distances obtained by the other methods.

In Table 5, we show the correlation of the original variables and the first principal component generated by the vertex, center, Best Point Distance, and Best Point Variance methods. For all variables except for $X_{6}$, the correlation was stronger with Best Point Distance or Best Point Variance. It can be inferred that the original variables were better represented by the PC1 component of the Best Point Distance and Best Point Variance methods. Interestingly, the correlation of the PC1 component from the Best Point Distance and Best Point Variance methods was stronger with the variables for the upper part of the face. The same result was generated if the correlations of the original variables were analyzed with respect to the other principal components of Best Point Distance and Best Point Variance.

## 4. Conclusions

This work focused on improving the center and vertex principal component methodology for interval-valued data. Compared with classical methods, symbolic methods based on interval-valued variables have important advantages, such as improved computational complexity due to reduced execution times, as small data tables are used. For example, for the facial recognition example, a table was passed from 27,000 cases to only 1 of 27 cases. In addition, symbolic methods allow for much better handling and interpretation of data variability. In the facial recognition scenario, the variation in the distances that measure different variables from one photo to another of the same person (such as the variation in the distance of the eyes $X_{1}$ from one photo to another) was due to the variation in the angle at which the photo was taken.

The Best Point methods proposed in this paper considerably improved both the center method and the vertex method. This is because Best Point Variance maximized the variance explained by the components and Best Point Distance minimized the squared distance between the vertices of the hyper-rectangles and their respective projections. As shown in the tables above, this led to a substantial improvement in all the quality indices used in principal component analysis. The result is better data clustering and, therefore, better prediction.

In future works, a consensus between the Best Point Variance and Best Point Distance methods could be constructed by applying a multiobjective optimization method to the functions φ and Λ. Finally, all the proposed algorithms for executing symbolic analyses of interval-valued data are available in the RSDA package in R (see [15]).

## Figures and Tables

**Figure 1 entropy-21-01016-f001:**
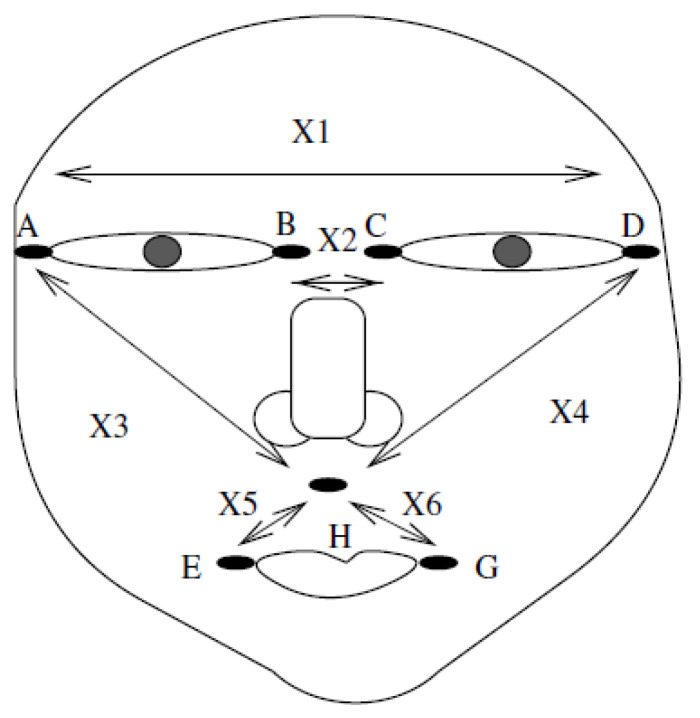
Random variables for facial description.

**Figure 2 entropy-21-01016-f002:**
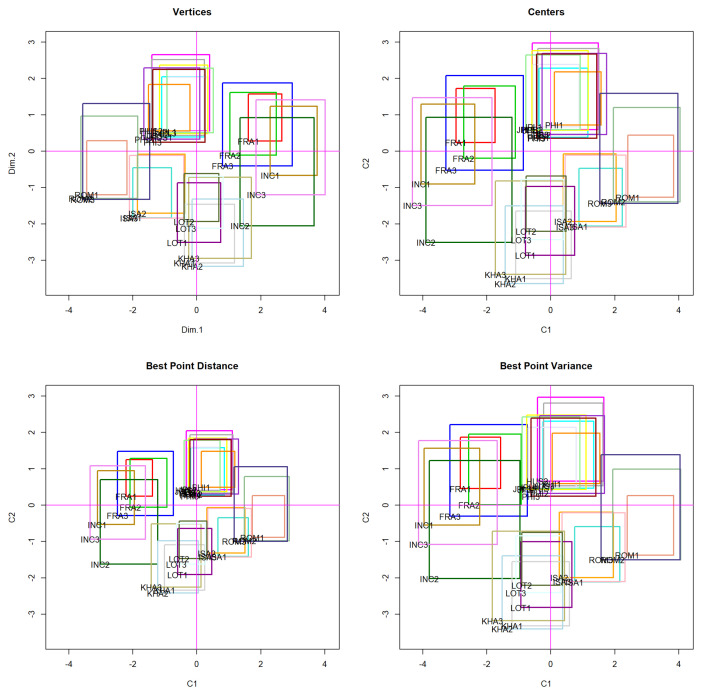
Principal component analysis (PCA) comparison.

**Table 1 entropy-21-01016-t001:** Faces data set.

Case	$X_{1}=$ AD	$X_{2}=$ BC	$X_{3}=$ AH	$X_{4}=$ DH	$X_{5}=$ EH	$X_{6}=$ GH
FRA1	[155.00, 157.00]	[58.00, 61.01]	[100.45, 103.28]	[105.00, 107.30]	[61.40, 65.73]	[64.20, 67.80]
FRA2	[154.00, 160.01]	[57.00, 64.00]	[101.98, 105.55]	[104.35, 107.30]	[60.88, 63.03]	[62.94, 66.47]
FRA3	[154.01, 161.00]	[57.00, 63.00]	[99.36, 105.65]	[101.04, 109.04]	[60.95, 65.60]	[60.42, 66.40]
HUS1	[168.9,172.84]	[58.55,63.39]	[102.83,106.53]	[122.38,124.52]	[56.73,61.07]	[60.44,64.54]
HUS2	[169.8,175.03]	[60.21,64.38]	[102.94,108.71]	[120.24,124.52]	[56.73,62.37]	[60.44,66.84]
HUS3	[168.8,175.15]	[61.4,63.51]	[104.35,107.45]	[120.93,125.18]	[57.2,61.72]	[58.14,67.08]
INC1	[155.3,160.45]	[53.15,60.21]	[95.88,98.49]	[91.68,94.37]	[62.48,66.22]	[58.9,63.13]
INC2	[156.3,161.31]	[51.09,60.07]	[95.77,99.36]	[91.21,96.83]	[54.92,64.2]	[54.41,61.55]
INC3	[154.5,160.31]	[55.08,59.03]	[93.54,98.98]	[90.43,96.43]	[59.03,65.86]	[55.97,65.8]
ISA1	[164,168]	[55.01,60.03]	[120.28,123.04]	[117.52,121.02]	[54.38,57.45]	[50.8,53.25]
ISA2	[163,170]	[54.04,59]	[118.8,123.04]	[116.67,120.24]	[55.47,58.67]	[52.43,55.23]
ISA3	[164,169.01]	[55,59.01]	[117.38,123.11]	[116.67,122.43]	[52.8,58.31]	[52.2,55.47]
JPL1	[167.1,171.19]	[61.03,65.01]	[118.23,121.82]	[108.3,111.2]	[63.89,67.88]	[57.28,60.83]
JPL2	[169.1,173.18]	[60.07,65.07]	[118.85,120.88]	[108.98,113.17]	[62.63,69.07]	[57.38,61.62]
JPL3	[169,170.11]	[59.01,65.01]	[115.88,121.38]	[110.34,112.49]	[61.72,68.25]	[59.46,62.94]
KHA1	[149.3,155.54]	[54.15,59.14]	[111.95,115.75]	[105.36,111.07]	[54.2,58.14]	[48.27,50.61]
KHA2	[149.3,155.32]	[52.04,58.22]	[111.2,113.22]	[105.36,111.07]	[53.71,58.14]	[49.41,52.8]
KHA3	[150.3,157.26]	[52.09,60.21]	[109.04,112.7]	[104.74,111.07]	[55.47,60.03]	[49.2,53.41]
LOT1	[152.6,157.62]	[51.35,56.22]	[116.73,119.67]	[114.62,117.41]	[55.44,59.55]	[53.01,56.6]
LOT2	[154.6,157.62]	[52.24,56.32]	[117.52,119.67]	[114.28,117.41]	[57.63,60.61]	[54.41,57.98]
LOT3	[154.8,157.81]	[50.36,55.23]	[117.59,119.75]	[114.04,116.83]	[56.64,61.07]	[55.23,57.8]
PHI1	[163.1,167.07]	[66.03,68.07]	[115.26,119.6]	[116.1,121.02]	[60.96,65.3]	[57.01,59.82]
PHI2	[164,168.03]	[65.03,68.12]	[114.55,119.6]	[115.26,120.97]	[60.96,67.27]	[55.32,61.52]
PHI3	[161,167]	[64.07,69.01]	[116.67,118.79]	[114.59,118.83]	[61.52,68.68]	[56.57,60.11]
ROM1	[167.2,171.24]	[64.07,68.07]	[123.75,126.59]	[122.92,126.37]	[51.22,54.64]	[49.65,53.71]
ROM2	[168.2,172.14]	[63.13,68.07]	[122.33,127.29]	[124.08,127.14]	[50.22,57.14]	[49.93,56.94]
ROM3	[167.1,171.19]	[63.13,68.03]	[121.62,126.57]	[122.58,127.78]	[49.41,57.28]	[50.99,60.46]

**Table 2 entropy-21-01016-t002:** Comparison of the first two principal components from the four methods.

	Vertex		Center		Best Point Distance		Best Point Variance	
**Cases**	**PC1**	**PC2**	**PC1**	**PC2**	**PC1**	**PC2**	**PC1**	**PC2**
FRA1	[1.61,2.66]	[0.27,1.57]	[-2.97,-1.75]	[0.24,1.72]	[−2.21,−1.39]	[0.24,1.25]	[−2.83,−1.57]	[0.45,1.87]
FRA2	[1.03,2.49]	[−0.11,1.61]	[−2.73,−1.11]	[−0.2,1.79]	[−2.09,−0.93]	[−0.06,1.29]	[−2.57,−0.92]	[0.01,1.95]
FRA3	[0.81,2.99]	[−0.4,1.88]	[−3.29,−0.86]	[−0.52,2.08]	[−2.49,−0.73]	[−0.29,1.48]	[−3.15,−0.73]	[−0.31,2.21]
HUS1	[−1.1,0.24]	[0.39,2.05]	[−0.37,1.16]	[0.38,2.28]	[−0.19,0.86]	[0.28,1.58]	[−0.24,1.34]	[0.47,2.31]
HUS2	[−1.41,0.4]	[0.56,2.65]	[−0.58,1.48]	[0.59,2.98]	[−0.33,1.11]	[0.42,2.04]	[−0.42,1.66]	[0.66,2.97]
HUS3	[−1.42,0.24]	[0.43,2.52]	[−0.4,1.51]	[0.46,2.82]	[−0.21,1.13]	[0.33,1.94]	[−0.22,1.63]	[0.53,2.8]
INC1	[2.29,3.77]	[−0.67,1.23]	[−4.07,−2.38]	[−0.9,1.29]	[−3.11,−1.95]	[−0.53,0.95]	[−3.97,−2.23]	[−0.55,1.56]
INC2	[1.35,3.66]	[−2.05,0.92]	[−3.91,−1.21]	[−2.51,0.93]	[−3.02,−1.22]	[−1.63,0.7]	[−3.8,−0.97]	[−2.02,1.22]
INC3	[1.86,4.02]	[−1.2,1.41]	[−4.33,−1.84]	[−1.5,1.47]	[−3.34,−1.61]	[−0.94,1.08]	[−4.14,−1.67]	[−1.08,1.77]
ISA1	[−2.01,−0.8]	[−1.83,−0.46]	[0.88,2.24]	[−2.06,−0.48]	[0.66,1.61]	[−1.42,−0.34]	[0.75,2.15]	[−2.11,−0.59]
ISA2	[−1.86,−0.37]	[−1.71,−0.08]	[0.39,2.04]	[−1.93,−0.07]	[0.32,1.51]	[−1.32,−0.07]	[0.26,1.95]	[−1.99,−0.19]
ISA3	[−2.11,−0.41]	[−1.84,−0.12]	[0.44,2.36]	[−2.09,−0.11]	[0.34,1.7]	[−1.43,−0.09]	[0.35,2.32]	[−2.1,−0.21]
JPL1	[−0.92,0.36]	[0.54,2.03]	[−0.57,0.89]	[0.67,2.38]	[−0.29,0.73]	[0.43,1.59]	[−0.7,0.79]	[0.49,2.14]
JPL2	[−1.17,0.34]	[0.48,2.37]	[−0.59,1.17]	[0.58,2.76]	[−0.25,0.93]	[0.37,1.84]	[−0.76,1.1]	[0.43,2.47]
JPL3	[−0.93,0.52]	[0.5,2.28]	[−0.78,0.92]	[0.59,2.64]	[−0.4,0.73]	[0.38,1.78]	[−0.89,0.91]	[0.45,2.43]
KHA1	[−0.39,1.18]	[−3.08,−1.46]	[−1.1,0.64]	[−3.51,−1.65]	[−1.01,0.25]	[−2.34,−1.09]	[−1.21,0.58]	[−3.32,−1.56]
KHA2	[−0.15,1.46]	[−3.17,−1.32]	[−1.43,0.39]	[−3.65,−1.5]	[−1.23,0.05]	[−2.42,−0.98]	[−1.52,0.37]	[−3.41,−1.39]
KHA3	[−0.25,1.71]	[−2.95,−0.72]	[−1.73,0.47]	[−3.39,−0.82]	[−1.42,0.13]	[−2.26,−0.52]	[−1.83,0.43]	[−3.18,−0.72]
LOT1	[−0.61,0.74]	[−2.51,−0.87]	[−0.79,0.74]	[−2.86,−0.98]	[−0.6,0.47]	[−1.91,−0.64]	[−0.94,0.66]	[−2.82,−1.01]
LOT2	[−0.4,0.69]	[−1.94,−0.62]	[−0.77,0.47]	[−2.2,−0.69]	[−0.55,0.31]	[−1.47,−0.44]	[−0.91,0.36]	[−2.21,−0.75]
LOT3	[−0.34,0.82]	[−2.12,−0.7]	[−0.93,0.41]	[−2.44,−0.78]	[−0.64,0.26]	[−1.63,−0.51]	[−1.09,0.33]	[−2.41,−0.85]
PHI1	[−1.51,−0.22]	[0.56,1.84]	[0.11,1.57]	[0.72,2.18]	[0.14,1.19]	[0.49,1.47]	[0.05,1.53]	[0.59,1.97]
PHI2	[−1.66,0.09]	[0.33,2.29]	[−0.27,1.74]	[0.45,2.69]	[−0.1,1.3]	[0.3,1.82]	[−0.35,1.67]	[0.32,2.46]
PHI3	[−1.38,0.25]	[0.25,2.25]	[−0.45,1.44]	[0.36,2.67]	[−0.22,1.07]	[0.25,1.8]	[−0.62,1.4]	[0.24,2.39]
ROM1	[−3.45,−2.19]	[−1.2,0.29]	[2.41,3.84]	[−1.27,0.44]	[1.74,2.74]	[−0.89,0.26]	[2.39,3.84]	[−1.38,0.26]
ROM2	[−3.63,−1.85]	[−1.3,0.97]	[1.96,4.04]	[−1.39,1.2]	[1.49,2.89]	[−0.98,0.79]	[1.94,4.06]	[−1.5,0.99]
ROM3	[−3.57,−1.48]	[−1.33,1.31]	[1.53,3.98]	[−1.44,1.58]	[1.17,2.83]	[−1,1.06]	[1.57,4.04]	[−1.5,1.39]

**Table 3 entropy-21-01016-t003:** Comparison of the variance resulting from different methods using facial recognition data.

	Vertex	Center	Best Point Distance	Best Point Variance
PC1	42.67%	46.47%	45.49%	56.01%
PC2	72.64%	80.53%	81.05%	88.31%
PC3	83.35%	89.65%	91.25%	99.72%
PC4	91.28%	95.06%	95.80%	99.85%
PC5	96.86%	98.96%	99.28%	99.97%
PC6	100.00%	100.00%	100.00%	100.00%

**Table 4 entropy-21-01016-t004:** Comparison of the distances resulting from different methods in facial recognition data.

	Vertex	Center	Best Point Distance	Best Point Variance
	10368.00	12719.64	6676.43	12457.09

**Table 5 entropy-21-01016-t005:** Correlation between the first component of each method and the variables.

	$X_{1}=$ AD	$X_{2}=$ BC	$X_{3}=$ AH	$X_{4}=$ DH	$X_{5}=$ EH	$X_{6}=$ GH
Vertex-PC1	0.64	0.49	0.84	0.89	−0.47	−0.43
Center-PC1	0.61	0.47	0.83	0.88	−0.52	−0.47
BestPointDistance-PC1	0.65	0.48	0.84	0.90	−0.46	−0.43
BestPointVariance-PC1	0.63	0.49	0.80	0.88	−0.55	−0.43

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
