# Peer review of "Optimized Dimensionality Reduction Methods for Interval-Valued Variables and Their Application to Facial Recognition"

_entropy, 2019, doi:10.3390/e21101016_

Round 1

Reviewer 1 Report

The paper presents two new methods to improve PCA of interval data. It is a honest paper, pretty well built, with an interesting example and with only minor English deficiencies. As such, it might be published nearly as it is.

Nevertheless, I have a remark which I would like to be taken into account by the authors: the reported improvements of the proposed methods, in respect to the already existing, do not appear so relevant to be considered systematic. Indeed, it is frequent that the proposal of a new method, or the improvement of an existing one, are accompanied by one example only. I do not agree with this habit and I suggest that at least a second example would be presented, in order to ensure that the better quality of the proposed methods is not episodic, unless they may show a theoretical proof.

In addition, I would be delighted seeing the faces of the example, as resulting from the PCA two-dimensional reconstruction, in comparison with some original image.

Author Response

Please see the file: Respond reviewer 1 - MDPI.pdf

Reviewer 2 Report

My main problem with current state of the paper is that I am not able to reproduce your results according your description. The main problem is that in Algorithm 1, the first step is "Apply PCA to Z" but I cannot understand with 100% certainty (from current paper and after reading [13]), HOW you exactly apply PCA as Z is 3 dimensional matrix. Is centers method use? Any other method? I cannot be sure. I think it would really benefit this paper if the code (in whatever language you used) would be linked in paper (in public repository).

Therefore, from Algorithm 1 onwards, I am just relaying on my "reading" and "comprehension" skills instead of being able to reproduce your results precisely - I think to have impact with this paper, any reader should be able to use it. I agree that the idea from the later part of the paper can be used but without total certainty I understood it correctly

Please check the following:

Line 90- too many times word next Formula 12 - something looks off Formula 19 - if I am not wrong, small zij is not defined (only Zij has been previously been defined) Are they the same? Centers method, PC1+PC2 should be 80.53? instead of 80.54? Theorem 1- z_up and z_low not defined in this paper? I am also slightly confused about some of the math. X is original data. Z, following line 54, seems to be scaled X. Therefore, formula after line 89 confuses me as Z is further scaled using Z? Maybe it is wrong but it goes back to the fact that zij is not defined exactly and I am having assumption zij=Zij as defined after line 54. Scaling by X and scaling by scaled Z, gives exactly same results. Therefore some of the math (for example, formula 22, confuses me)

The application part with face recognition dataset is good but as I said - I miss the ability to test it myself using your coded algorithm. (I was able to reproduce given results using your dataset for vertex and centers method). If the results are indeed as good, as with best point variance, then this method can be really beneficial.

Another aspect, not covered, is that by the looks of it, this proposed method suffers from the same drawback as vertex method - all combinations of vertex must be produced. The computational limitations have not been covered.

Author Response

Please see the file: Respond reviewer 2 - MDPI.pdf
